# Photoswitchable dynamic conjugate addition-elimination reactions as a tool for light-mediated click and clip chemistry

Hanwei Lu [1,2], Hebo Ye[1], Meilan Zhang[1], Zimu Liu[1], Hanxun Zou[1] & Lei You [1,2,3] ✉

Phototriggered click and clip reactions can endow chemical processes with high spatiotemporal resolution and sustainability, but are challenging with a limited scope. Herein we report photoswitchable reversible covalent conjugate addition-elimination reactions toward light-addressed modular covalent connection and disconnection. By coupling between photochromic dithienylethene switch and Michael acceptors, the reactivity of Michael reactions was tuned through closed-ring and open-ring forms of dithienylethene, allowing switching on and off dynamic exchange of a wide scope of thiol and amine nucleophiles. The breaking of antiaromaticity in transition states and enol intermediates of addition-elimination reactions provides the driving force for photoinduced change in kinetic barriers. To showcase the versatile application, light-mediated modification of solid surfaces, regulation of amphiphilic assemblies, and creation/degradation of covalent polymers on demand were achieved. The manipulation of dynamic click/clip reactions with light should set the stage for future endeavors, including responsive assemblies, biological delivery, and intelligent materials.

Through reversible formation, breakage, and exchange of covalent bonds dynamic covalent chemistry (DCC) has the advantage of adaptivity, self-correction, and degradability, with broad utility in the creation of functional molecules, assemblies, and materials[1–5]. Click chemistry allows modular covalent connection and has become one powerful tool in chemical, biological, and materials sciences[6–10]. To the contrary, clip (or declick) reactions focus on efficient bond cleavage[11,12]. Covalent connection and disconnection via DCC contributes to the development of both click and clip chemistry in a reversible and recyclable manner[13–16]. Light can serve as a clean stimulus for actively regulating chemical processes with high spatiotemporal resolution, giving access to high energy species and attracting extensive attention[17–20]. Photoswitches can bidirectionally shift between a thermodynamically stable state and a kinetically trapped metastable state in response to light and represent an idea way for

driving DCC[21–23]. By coupling between a photoswitch and a dynamic covalent reaction (DCR) light-gated reactivity provides a layer of control unattainable in thermal reactions and further offers the dual benefits of light-mediated bond making and breaking. Through masking/unmasking reactive olefin and carbonyl functional groups with diarylethene, photoswitchable Diels–Alder reactions[24–26] and imine condensation/hydrolysis[27] were realized, respectively. Moreover, azobenzene and acylhydrazone attached boronic acids enabled tuning the equilibria and kinetics of cyclic boronic ester exchange by utilizing light-modulated intramolecular hydrogen bonding[28,29]. Recently, we reported remote control over dynamic C-N, C-O, and C-S bonds for switchable stability/degradability in covalent architectures via photoinduced ring-chain tautomerism and dual reactivity based DCC[30]. Despite significant advances, photoswitchable DCC remains challenging with limited scope.

[1]State Key Laboratory of Structural Chemistry, Fujian Institute of Research on the Structure of Matter, Chinese Academy of Sciences, 350002 Fuzhou, Fujian, China. [2]University of Chinese Academy of Sciences, 100049 Beijing, China. [3]Fujian Science & Technology Innovation Laboratory for Optoelectronic Information of China, 350002 Fuzhou, Fujian, China. ✉e-mail: lyou@fjirsm.ac.cn

Michael-type reactions are one class of the most explored among an expanding toolbox of DCRs and reversible click reactions (Fig. 1A)[31–34]. The synthetic accessibility and structural diversity of mono- and di-activated Michael acceptors, including α,β-unsaturated carbonyls and associated scaffolds (a of Fig. 1A), offer ample means for manipulating thermodynamic and kinetic features of conjugate additions. The introduction of a leaving group (XR$^3$, X = O, S, NR) at β position further allows the realization of dynamic conjugate addition-elimination reactions toward the exchange of nucleophiles (b of Fig. 1A), thus providing opportunities for bond click and clip[35–38]. Dynamic conjugate additions were employed for the identification of protein inhibitors[39–41] and the assembly of nanoparticles[42,43]. Moreover, Michael reactions afford a versatile avenue for the construction of covalent organic frameworks[44–46] and adaptable polymer networks[47–51]. In addition, the creation of degradable covalent polymers was achieved by taking advantage of addition-elimination reactions of 1,3-dicarbonyl derived conjugate acceptors[52–54]. Light-induced thiol-ene click chemistry is complementary with Michael reactions, albeit through photoinitiated radical mechanism[55,56]. Photoswitchable dynamic conjugate addition-elimination reactions are rare, to the best of our knowledge. The efficient control of diverse DCRs with light would be advantageous to obtain new structures and functions.

Taking into consideration of the reaction mechanism (Fig. 1A), we were wondering the possibility of stabilizing/destabilizing Michael adduct intermediate (i.e., enol)[57,58] with light for accelerating/decelerating the reaction. Toward this end, the conjugate acceptor of 4-cyclopentene-1,3-diketone derivative was fused with dithienylethene (DTE)[23] on the ethene bridge (Fig. 1B). Cyclic 1,3-diketone scaffolds play a notable role in organic synthesis, medicinal chemistry, and optoelectronic materials[59–61]. We postulated that the rearrangement of π systems of DTE unit would induce a change in the reactivity of the Michael acceptor. Upon bidirectional photoswitching for photocyclization/photocycloreversion of DTE the open-ring and closed-ring forms would afford Michael adduct intermediates with 4π and 14π electrons, respectively. The varying antiaromatic character of these enol species and associated transition states would in turn influence the kinetic barriers of addition-elimination reactions, hence turning off/on the reactivity and achieving photoswitchable exchange of nucleophiles for covalent bond formation and scission. In the current work, we report light-induced dynamic conjugate addition-elimination reactions along with associated mechanistic insights. The application in the regulation of molecular assemblies and covalent polymers was further demonstrated, enriching the toolbox of reversible click/clip chemistry and paving the way for future studies.

## Results and discussion
### Design and synthesis
To realized the concept, a conjugate acceptor, namely 2-(ethoxymethylene)−4-cyclopentene-1,3-diketone (**1**), was chosen as the reactive motif with photoswitchble DTE unit attached at 4,5-position (Fig. 2A). The conjugate addition of mononucleophiles, such as thiols and amines, would lead to the formation of tetrahedral intermediate

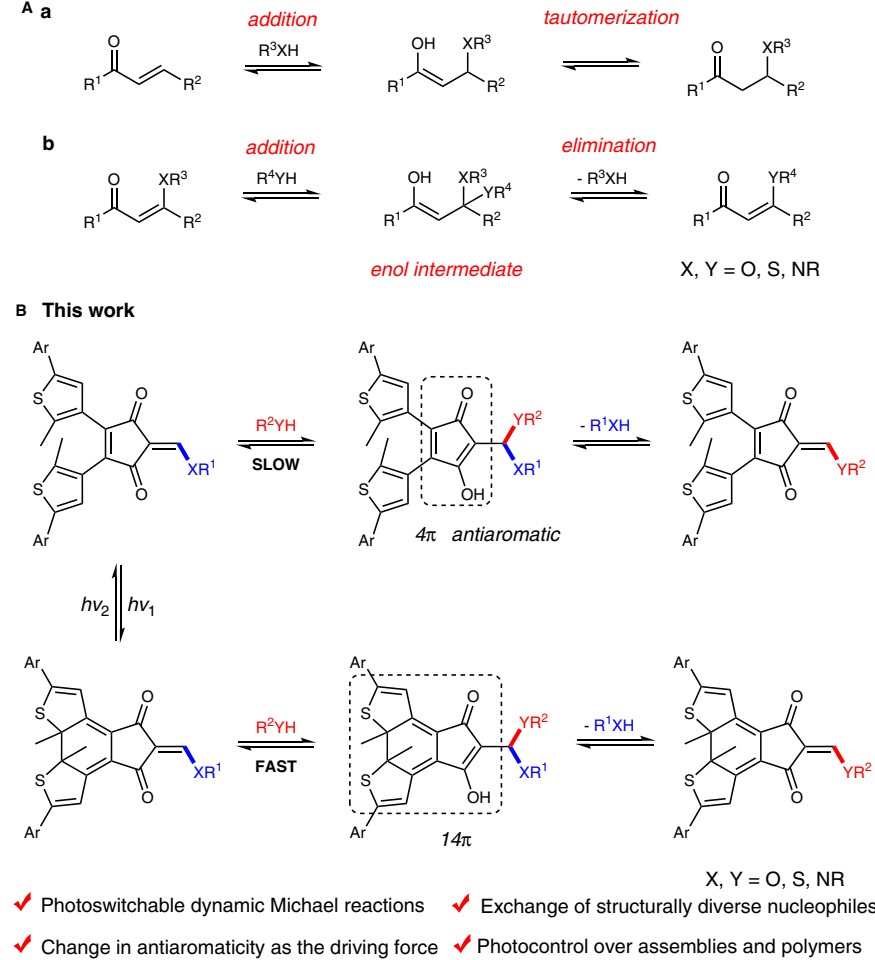

**Fig. 1 | The concept of photoswitchable dynamic conjugate addition-elimination reactions. A** Conjugate addition reactions (a) and addition-elimination reactions (b), with associated enol intermediates shown. **B** This work of photoswitchable dynamic conjugate addition-elimination reactions for light-mediated covalent connection (in red) and disconnection (in blue), with the change in antiaromaticity as the driving force.

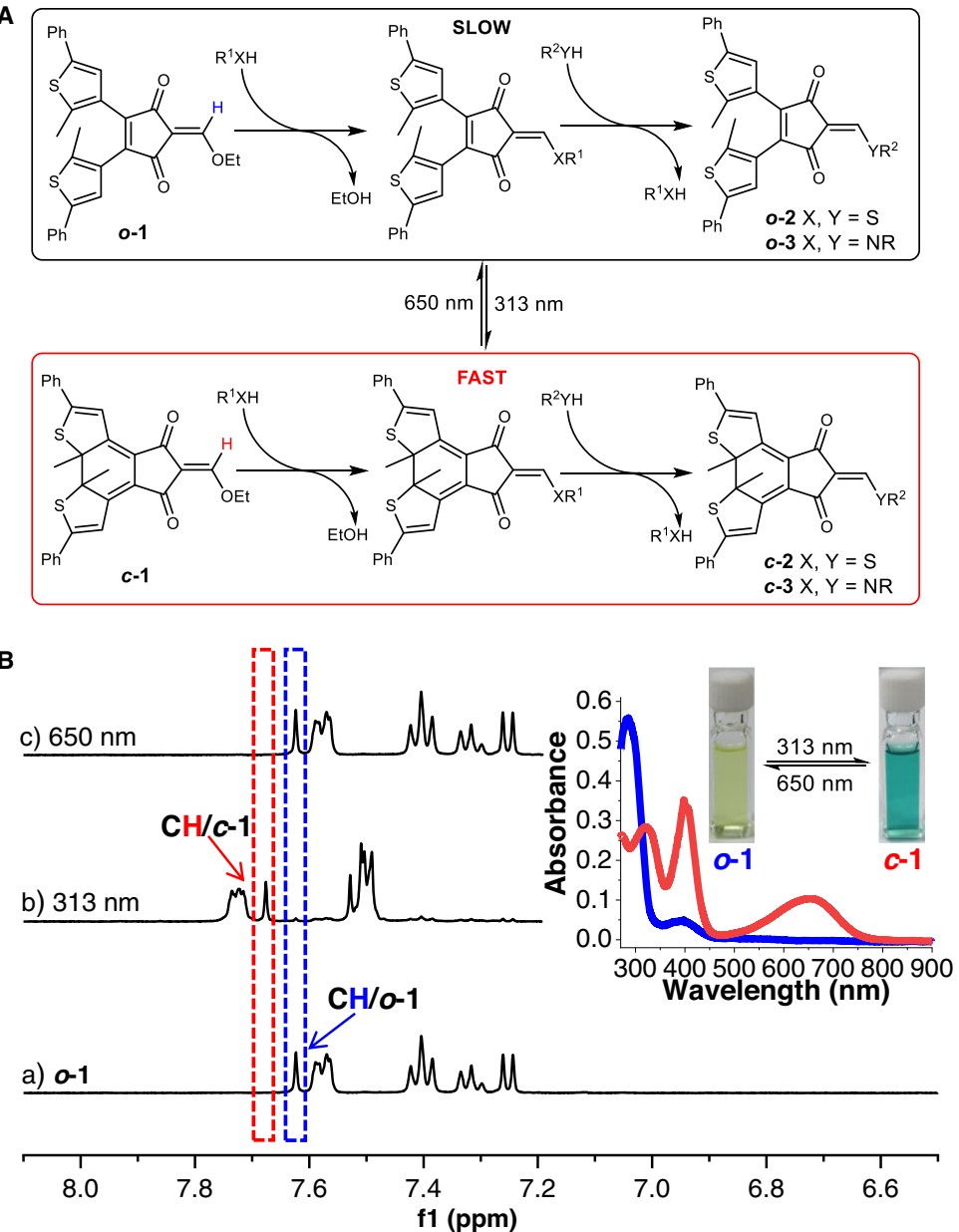

**Fig. 2 | Structures and photochromic behaviors of conjugate acceptors.**
**A** Photoswitchable dynamic conjugate addition-elimination reactions between
*o−1/c−1* and amine/thiol nucleophiles as well as the subsequent exchange reactions.
**B** Photoswitching between *o-1* and *c−1* (5 mM) upon irradiation at 313 and 650 nm
monitored by [1]H NMR (400 MHz, 20 °C) in CD$_3$CN, with UV-vis spectra and pho-
tographs of *o-1* and *c-1* (25 μM in CH$_3$CN) in the inset.

thioacetals and hemiaminal ethers, respectively. The subsequent
departure of alcohols would give enol thioethers (**2**) and enamines (**3**),
with each again as a new conjugate acceptor susceptible to Michael/
retro-Michael reactions for the exchange of nucleophiles. In short, we
sought a versatile platform wherein dynamic conjugate addition-
elimination reactions would be dictated by light in conjunction with
varying reactivity and departing ability of diverse nucleophiles.

DTE modified cyclopentene-1,3-diketone was constructed
through Wittig–Horner reaction of 2,3-dithienylmaleic anhydride and
phosphorus ylide followed by the rearrangement (Supplementary
Figs. 1 and 2). The condensation reaction between 1,3-diketone and
triethyl orthoformate afforded the desired conjugate acceptor (*o-1*,
Supplementary Fig. 3). Upon electrocyclic ring-closure at 313 nm for
1.5 h, *c-1* was obtained in high yield (95%) in the photostationary state
(Fig. 2B). The irradiation at 650 nm (2 h) gave *o-1* again, accomplishing
bidirectional photoswitching. Furthermore, multiple cycles of

photoswitching were realized, exhibiting great reversibility and
fatigue-resistance (Supplementary Fig. 29). The measurement of
absorbance spectra validated photochromic behaviors, as evidenced
by yellow and green color for *o-1* and *c-1* (Fig. 2B and Supplementary
Figs. 30 and 31).

### Light-controlled exchange of alcohol/thiol nucleophiles

With the strategy in place, the kinetics of conjugate addition-
elimination reactions was tracked. To contrast the reactivity directly,
the individual reactions of *o−1* and *c−1* with nucleophiles were run. For
example, the equilibrium of the reaction of *c-1* with 1-propanol was
reached after 60 h, but *o-1* remained nearly unchanged after 3 weeks
(Supplementary Figs. 32–34, entry 1 of Table 1). Similarly, the reaction
of *o-1* with 1-propanethiol (3.0 equiv.) in CD$_3$CN proceeded pretty
slowly, affording a yield of 38% for enol thioether *o-2* after 35 days
(Fig. 3A). In contrast, it took 12 h for the reaction of *c-1* with

**Table 1 | Summary of exchange reactions between photoswitchable Michael acceptors and various nucleophilic reagents tracked by $^1$H NMR[a,b]**

1: X, Y = O    2: X, Y = S    3: X, Y = NR

| Entry | R$^1$XH | R$^2$YH | Open form | | Closed form | |
|---|---|---|---|---|---|---|
| | | | Time | Yield | Time | Yield |
| 1 | Ethanol | 1-Propanol | 21 d | n.r | 60 h | 37 % |
| 2 | Ethanol | 1-Propanethiol | 35 d | 38% | 12 h | 100% |
| 3 | Ethanol | p-Toluenethiol | 7 d | n.r | 27 h | 95% |
| 4 | Ethanol | 2-Propanethiol | 30 d | n.r | 50 h | 95% |
| 5 | Ethanol | t-Butylmercaptan | 45 d | n.r | 18 d | 76% |
| 6 | Ethanol | L-cysteine | 36 h | 100%[c] | 3 min | 100%[c] |
| 7 | 1-Propanethiol | Benzylthiol | 12 d | 48% | 2 h | 48% |
| 8 | Ethanol | 1-Butylamine | 3 min | 100% | 3 min | 100% |
| 9 | Ethanol | Isobutylamine | 3 min | 100% | 3 min | 100% |
| 10 | Ethanol | Aniline | 26 h | 100% | 3 min | 100% |
| 11 | Ethanol | p-Trifluromethylaniline | 15 d | 100% | 3 min | 100% |
| 12 | Ethanol | p-Anisidine | 12 h | 100% | 3 min | 100% |
| 13 | Ethanol | L-lysine | 3 min | 100%[c] | 3 min | 100%[c] |
| 14 | 1-Propanethiol | 1-Butylamine | 3 min | 100% | 3 min | 100% |
| 15 | 1-Propanethiol | Aniline | 9 d | 33% | 3 min | 100% |
| 16 | L-cysteine | L-lysine | 10 h | 100%[c] | 3 min | 100%[c] |
| 17 | Ethanol | Diisopropylamine | 7 d | 80% | 3 min | 100% |
| 18 | Ethanol | Piperidine | 3 min | 100% | 3 min | 100% |
| 19 | 1-Propanethiol | Diisopropylamine | 21 d | n.r | 20 h | 90 % |
| 20 | 1-Propanethiol | Piperidine | 3 min | 100% | 3 min | 100% |
| 21 | Isobutylamine | 1-Butylamine | 46 d | 6% | 4 d | 53% |
| 22 | Piperidine | 1-Butylamine | 10 d | 95% | 3 min | 100% |
| 23 | Aniline | 1-Butylamine | 37 d | 39% | 3 min | 100% |

[a]See detailed conditions in "Methods" and Supplementary Information.
[b]**2** and **3** were created in situ from reactions of **1** with thiols and amines, respectively.
[c]Reactions in a solvent mixture of CD$_3$CN:D$_2$O = 4:1 (0.5 mL), with others in CD$_3$CN (0.5 mL).

1-propanethiol for complete formation of **c-2** (Supplementary Figs. 35–38, entry 2 of Table 1). A mixture of **o-1** and thiol was also monitored under 313 nm illumination, faster than the reaction of **c-1** in the dark (Supplementary Figs. 39 and 40). To showcase the generality, a wide range of thiols, including aromatic thiols, was tested (Supplementary Figs. 42–53, entries 3-6 of Table 1). The reaction of **c−1** with p-toluenethiol gave desired **c-2** (95%), while **o−1** remained virtually intact after 1 week (Supplementary Figs. 42–44). Analogous results were obtained with sterically bulky 2-propanethiol and t-butanethiol, turning on and off their reactions with **c−1** and **o-1**, respectively (Supplementary Figs. 45–50). The successful covalent bonding of thiol containing amino acid (i.e., cysteine) with **c-1** was realized quantitatively (3 min) in aqueous media, further broadening the scope (Supplementary Figs. 51–53, entry 6 of Table 1). Moreover, photocycloreversion of **c-2** under 650 nm light afforded **o-2**, and bidirectional photoswitching between **o-2** and **c-2** for multiple cycles was successful with high efficiency and fatigue-resistance (Supplementary Fig. 41). Despite the sluggishness of the direct conversion from **o-1** to **o-2**, **o-2** was facilely obtained through a detour of reactive **c-1** and then **c-2**.

Having realized successful incorporation of thiols, the exchange of thiols was then studied. **c-2** incorporating 1-propanethiol was created followed by the addition of benzylthiol (Supplementary Figs. 54–56, entry 7 of Table 1). After reaching equilibrium readily after 2 h benzylthiol derived product accumulated, with a decrease in the amount of original **c-2** from 1-propanethiol. However, the corresponding thiol exchange with open-ring form **o-2** was kinetically disfavored. Such a difference between **o-2** and **c-2** for thiol exchange is consistent with the trend of reactions of **o-1** and **c-1** with thiols.

### Light-controlled exchange of amine nucleophiles

Encouraged by photocontrolled modulation of reactivity toward thiols, reactions with amines were next examined (Supplementary Figs. 57–104, entries 8-23). A mixture of **1** and 1-butylamine (1.5 equiv.) gave enamine **3** within 3 min in $^1$H NMR analysis irrespective of electrocyclic open-ring (**o−1**) and closed-ring (**c−1**) forms (Supplementary Figs. 57–59, entry 8 of Table 1). The successful creation of **o-3** was further validated by its crystal structure (Supplementary Fig. 28). DTE unit adopts an antiparallel conformation, and an exocyclic enamine

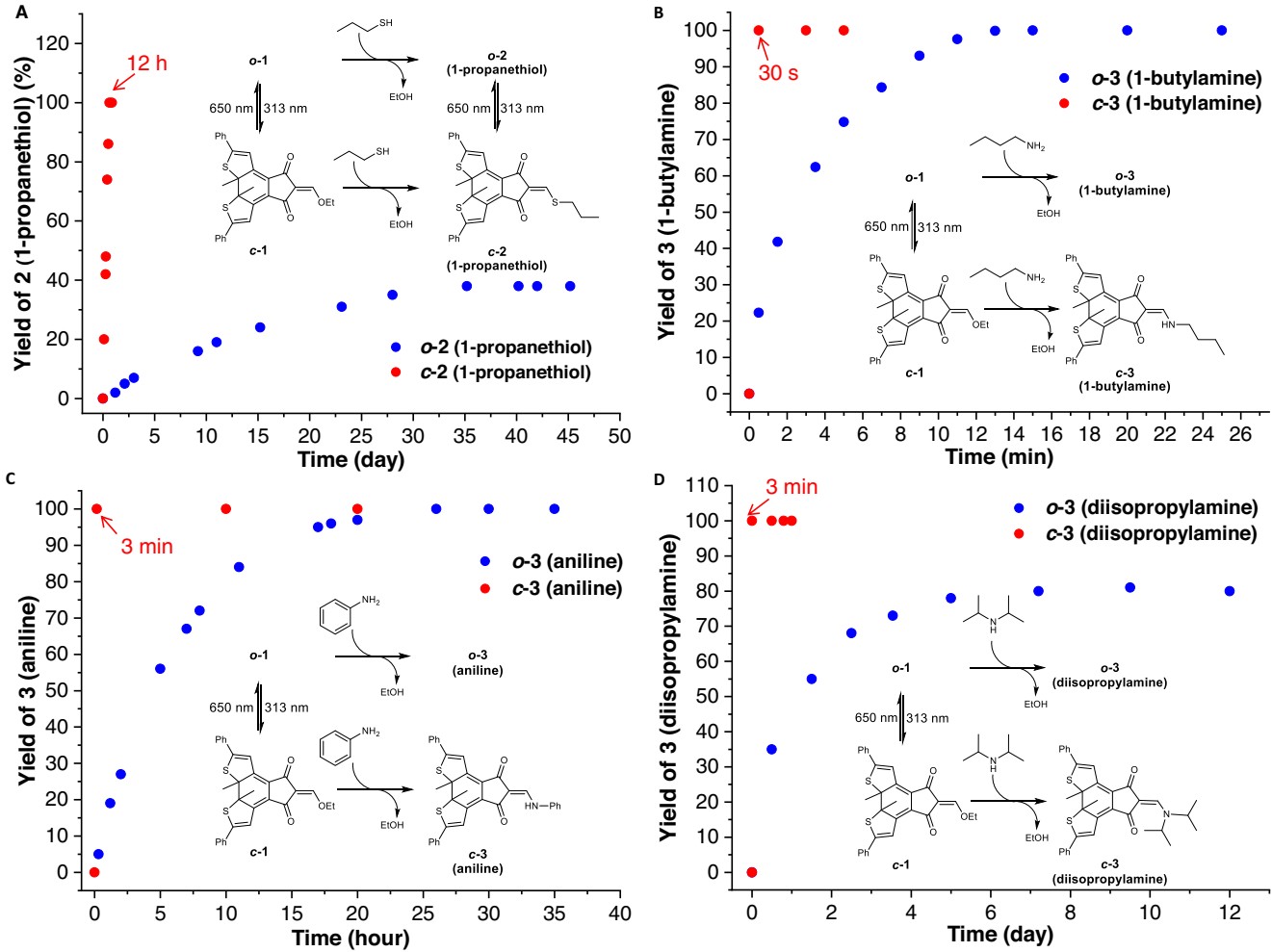

**Fig. 3 | Light-controlled conjugate addition-elimination reactions. A** Kinetics profile of the reaction of *o-1/c-1* (5 mM) and 1-propanethiol (3 equiv.) monitored by [1]H NMR in CD₃CN. **B** Kinetics profile of the reaction of *o-1/c-1* (25 μM) and 1-butylamine (1.5 equiv.) monitored by absorbance spectra in CH₃CN. **C** Kinetics profile of the reaction of *o-1/c-1* (5 mM) and aniline (1.5 equiv.) monitored by [1]H NMR in CD₃CN. **D** Kinetics profile of the reaction of *o-1/c-1* (5 mM) and diisopropylamine (1.5 equiv.) monitored by [1]H NMR in CD₃CN.

resides in cyclic 1,3-diketone motif, which is stabilized via intramolecular NH…O hydrogen bonding (2.20 Å). Analogous results were found with isobutylamine (Supplementary Figs. 60–62, entry 9 of Table 1). To probe the discrimination between *o−1* and *c−1* the kinetics was monitored by absorbance spectra in dilute solution. The reaction of *c-1* (25 μM) with 1-butylamine occurred immediately (within 30 s) once mixed together, while it took 13 min for the corresponding reaction of *o-1* to complete, indicating improved reactivity for closed-ring form (Fig. 3B and Supplementary Figs. 63 and 64).

Gratifyingly, the reactions of *o-1* and *c-1* with aniline (1.5 equiv.) required 26 h and 3 min to complete, respectively (Fig. 3C and Supplementary Figs. 65–67, entry 10 of Table 1). Through tracking the kinetics of *o-1* and *c-1* via [1]H NMR and UV-vis spectra (Supplementary Fig. 68), the rate enhancement was estimated be over 10⁴. The significantly enhanced reactivity of *c-1* over *o-1* further enabled its rapid and quantitative reactions with *p*-anisidine and *p*-trifluromethylaniline (Supplementary Figs. 69–74, entries 11 and 12 of Table 1). Moreover, the scission of C-S bond was switched off and on with the reactions of *o−2* and *c-2* with aniline, respectively (Supplementary Figs. 80–82, entry 15 of Table 1). Photoswitchable formation of **3** incorporating lysine was realized with cysteine derived **2** via amine/thiol exchange, and again strong acceleration was found with closed-ring form (Supplementary Figs. 83–85, entry 16 of Table 1).

The differentiation between *o−1* and *c-1* for sterically congested secondary amines was also feasible, further showcasing the power of the current system (Supplementary Figs. 86–92, entries 17 and 18 of Table 1). While the reaction of *c-1* and diisopropylamine (1.5 equiv.) led to quantitative formation of enamine *c-3* after 3 min, the corresponding reaction of *o-1* gave a yield of 80% (*o-3*) after 7 days (Fig. 3D and Supplementary Figs. 86–89). In addition, the reactions of *o−2* and *c-2* with secondary amines, such as diisopropylamine, for amine/thiol exchange were studied, kinetically favoring the latter (Supplementary Figs. 93–96, entries 19-20 of Table 1). In spite of electrocyclic ring-opening of *c-3* to form *o-3*, attempts to convert *o-3* back to *c-3* failed. This is likely due to photoinduced electron transfer (PET) imparted by nitrogen lone pair, resulting in the deactivation of excited state of *o-3*[62].

To explore photoinduced scission of C-N bonds amine/amine exchange was next investigated. The individual solutions of *o-3* and *c-3* incorporating isobutylamine created in situ were mixed with 1-butylamine and then tracked. The scrambling of amines was detected with *c-3* reaching equilibrium after 4 days, while *o-3* remained poorly reactive after 40 days (Supplementary Figs. 97, 98, entry 21 of Table 1). Furthermore, amine exchange took place within 3 min for the reaction of piperidine or aniline derived *c-3* with 1-butylamine, incorporating 1-butylamine quantitatively and matching the higher nucleophilicity of aliphatic primary amines. In contrast, the displacement of amines was

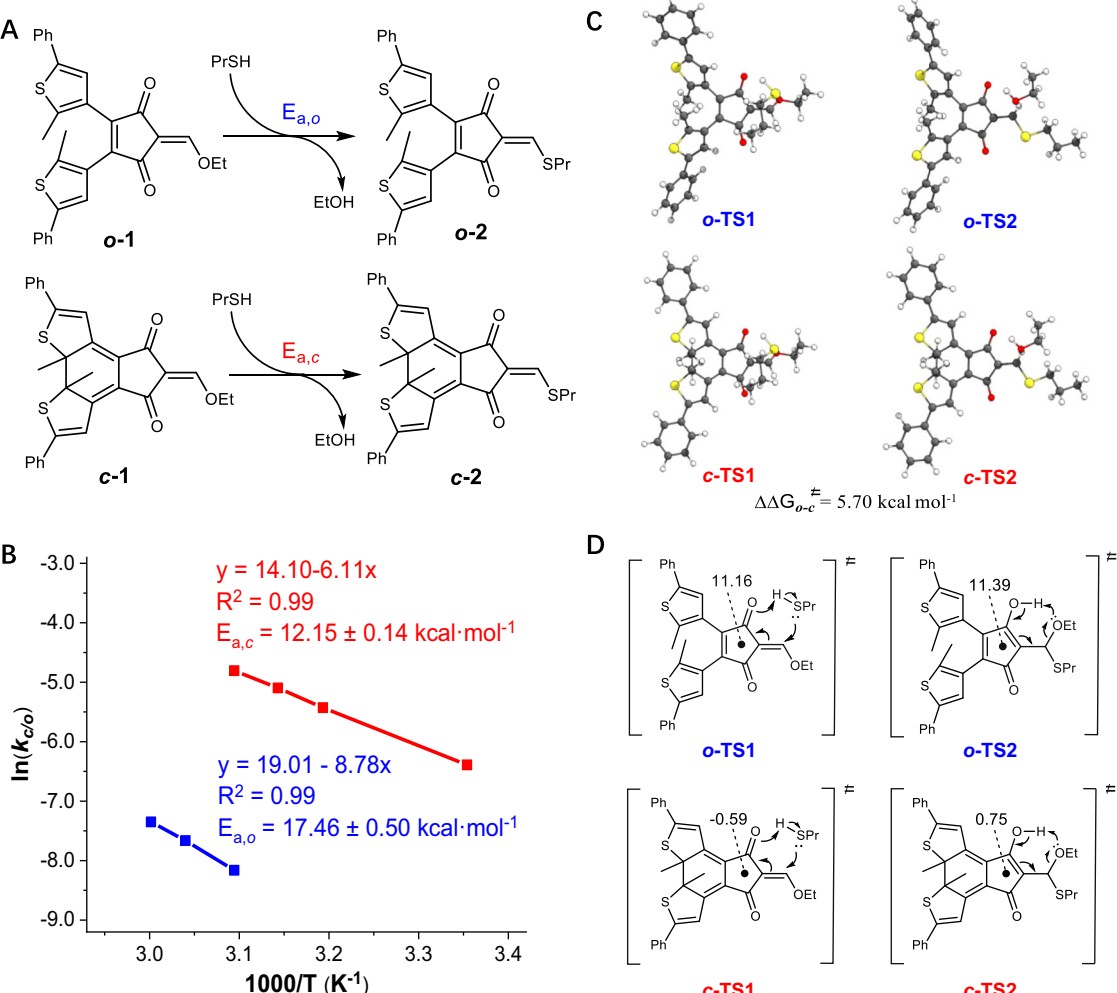

**Fig. 4 | Mechanistic insights via experimental and computational studies.** **A** Model reactions used for kinetics studies. **B** Arrhenius plots of exchanged reactions of *o*-1/*c*-1 with 1-propanethiol. **C** Calculated structures of *o*-TS1/*c*-TS1 and *o*-TS2/*c*-TS2 in acetonitrile, with the difference in relative free energies (kcal·mol⁻¹) listed. **D** NICS(1)$_{zz}$ (ppm) values of *o*-TS1/*c*-TS1 and *o*-TS2/*c*-TS2. The functional of M06−2X-D3 and the basis set of def2-SVPP with a polarizable continuum model (PCM) for acetonitrile were used.

again kinetically suppressed with corresponding *o*-3 (Supplementary Figs. 99–104, entries 22 and 23 of Table 1). As a result, the extent of acceleration for dynamic exchange reactions and the selectivity toward different types of nucleophiles can be regulated by light, enabling phototriggered covalent connection and disconnection.

## Mechanistic insights

To determine the activation energies of the addition-elimination reactions experimentally, the kinetics of *c*-1/*o*-1 with 1-propanthiol (10 equiv.) was studied at different temperatures (Fig. 4A). By fitting the experimental data with a pesudo-first order kinetics equation, the rate constants $k_{c/o}$ were obtained and then plotted in an Arrhenius plot (Supplementary Figs. 105 and 106), giving the activation energies of 17.46 kcal mol⁻¹ for *o*-2 and 12.15 kcal mol⁻¹ for *c*-2 (Fig. 4B), respectively. Furthermore, the rate constants of reactions of *c*-1/*o*-1 with aniline at room temperature were measured, modulating the rate over 4 orders of magnitude (Supplementary Fig. 107).

To probe the mechanism of light-activated DCRs, density functional theory (DFT) calculations were further employed to identify potential intermediates and transition states (TSs). Taking *o*-1 as an example, a concerted mechanism for nucleophilic attack of 1-propanethiol and proton transfer was found via a six-membered ring TS (*o*-TS1, Fig. 4C and Supplementary Fig. 108), resulting in thioacetal

(*o*-4) with a cyclic enol structure. The elimination of ethanol also proceeds through a six-membered ring TS (*o*-TS2) to afford enol thioether product *o*-2. Upon photocyclization of *o*-1 to create *c*-1 the energy is significantly raised. Similar to *o*-1, the combination of *c*-1 and 1-propanethiol gives thioacetal (*c*-4) via a six-membered ring TS (*c*-TS1), as also the case for subsequent departure of ethanol (*c*-TS2) to furnish *c*-2 (Fig. 4C and Supplementary Fig. 108). The kinetic barriers (ΔG‡) for addition and elimination sequences were compared, and the reaction of higher energy metastable form (i.e., *c*-1) is significantly kinetically favored over the thermodynamic stable form (*o*-1) by around 5.7 kcal mol⁻¹ (Fig. 4C), largely matching the difference in measured reaction barriers (5.3 kcal mol⁻¹).

The photoinduced activation is attributed to the loss of the antiaromatic feature for the stabilization of transition states (*c*-TS1 and *c*-TS2) as well as the enol intermediate (*c*-4) in their corresponding closed-ring forms (Fig. 4D and Supplementary Fig. 108). The calculation of nucleus-independent chemical shift (NICS) values[63], an index for quantifying aromaticity/antiaromaticity, confirmed the rationalization. *o*-4 afforded a NICS(1)$_{zz}$ value of 17.53 ppm, falling in line with its antiaromatic character. In contrast, a NICS(1)$_{zz}$ value of -0.25 ppm was found for *c*-4. Such a sharp decrease in NICS(1)$_{zz}$ echoes a loss of antiaromaticity in the enol form (*c*-4) upon photocyclization. Moreover, a similar trend was obtained with *o*-TS1/*c*-TS1 and *o*-TS2/*c*-TS2.

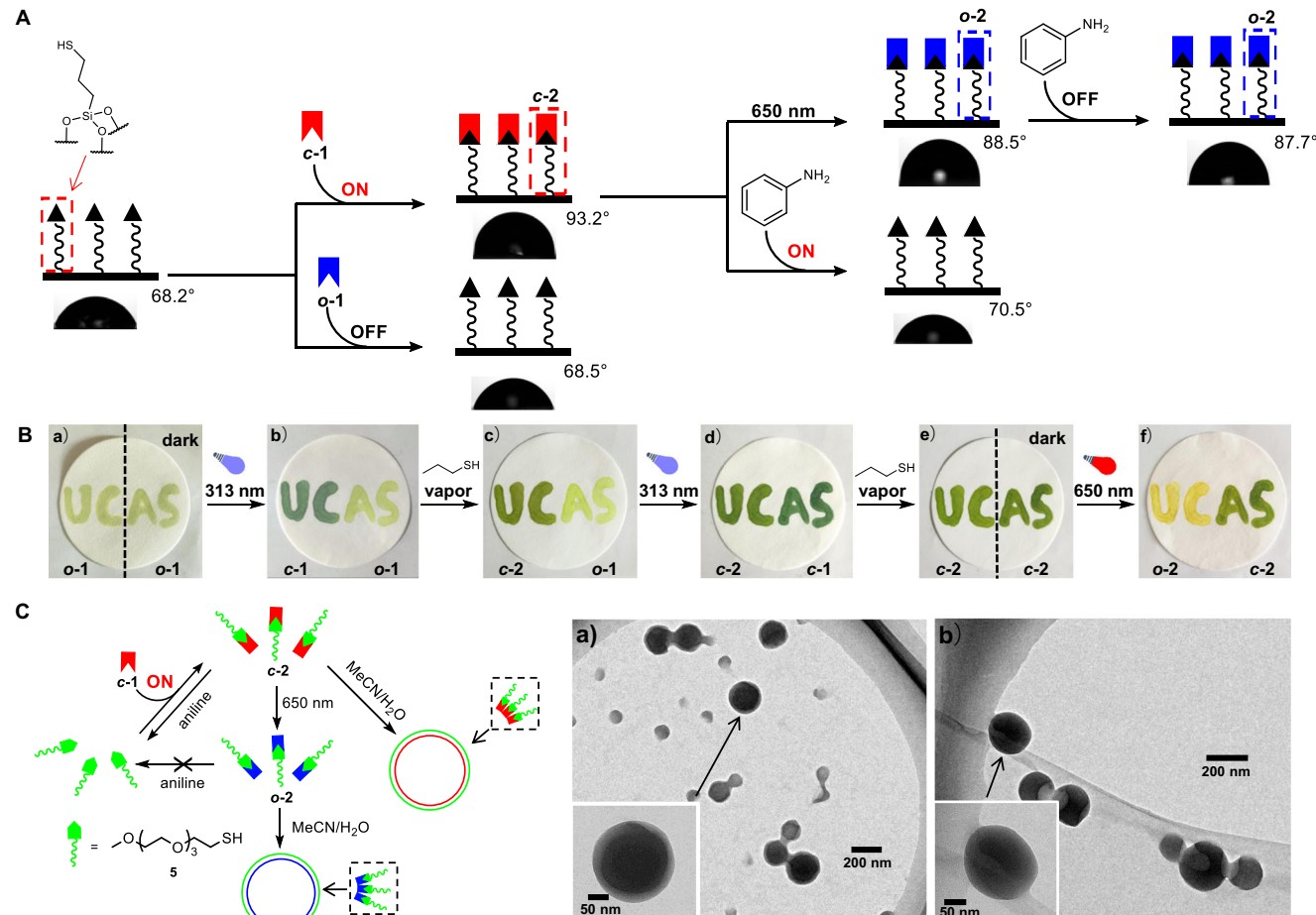

**Fig. 5 | Light-controlled regulation of assemblies. A** Light-mediated modification of solid surfaces, with surface contact angle shown. **B** Light-mediated modification of patterning. **C** Light-mediated modulation of amphiphilic assemblies. TEM images of the amphiphilic assemblies formed by **c−2** (a) and **o−2** after irradiation of **c−2** at 650 nm for 2 h (b).

Light-induced breaking of antiaromaticity plays a stabilizing role in **c-TS1** (−0.59 ppm) and **c-TS2** (0.75 ppm) as compared to **o-TS1** (11.16 ppm) and **o-TS2** (11.39 ppm). Analogous findings were found for the reaction of **1** with 1-butylamine, with a significantly lower barrier for **c-1** than **o-1** (Supplementary Fig. 109).

### Regulation of molecular assemblies

Having achieved photoswitchable conjugate addition-elimination reactions, we next set to show the application potential in the control of molecular assemblies. First, photoaddressed modification of solid surfaces was explored, as surfaces exhibiting switchable wettability and functionality could generate interest in many aspects, such as microfluidic chips and lithography[64–66]. Toward this end, glass slides with porous silica coating were modified with 3-mercaptopropyltriethoxysilane to introduce thiol groups (Fig. 5A). By immersing the glass into a solution of **c-1**, **c-2** was readily grated onto the surface via the formation of C-S bond, with the static water contact angle changing from 68.2° to 93.2°. In sharp contrast, the slide upon exposure with **o-1** remained nearly intact, as evidenced by the maintenance of the contact angle (68.5°). Moreover, the treatment of modified coating (**c-2**) with aniline solution led to the cleavage of C-S bond via amine/thiol exchange, thereby recovering the original state (contact angle from 93.2° to 70.5°). However, upon irradiation with 650 nm light to afford **o-2** (88.5°) the immobilization of attachment was apparent after the treatment with aniline (contact angle 87.7°), as the dynamic exchange reaction was turned off. In essence, photoswitchable modification of solid surfaces was achieved by utilizing active and inactive forms of conjugate acceptors for reversible click/clip reactions.

To further showcase the spatial resolution of light, light-induced modification of patterning was performed. The light yellow color of letters "UCAS" was written on a piece of filter paper with the solution of **o-1** (Fig. 5B). After illumination at 313 nm with "AS" masked in the dark, the color of "UC" turned blackish green due to the appearance of **c-1**. Since the exchange rate between **c-1/o-1** and thiols was different, letters "UC" selectively changed to green under 1-propanethiol vapor, with "AS" remaining intact. Upon irradiation at 313 nm and retreatment with 1-propanethiol vapor, "AS" was also written in the form **c-2**. The golden color of "UC" was finally obtained on purpose after illumination at 650 nm while "AS" was shielded from light. Photoswitchable reactivity thereby allowed the realization of the color-changing pattern with spatial resolution.

Light-controlled amphiphilic assemblies were next studied to demonstrate the versatility. An amphiphile (**c-2**) was synthesized in situ from **c-1** and a thiol containing a hydrophilic oligo(ethylene glycol) chain (**5**), with DTE serving as the hydrophobic portion (Fig. 5C and Supplementary Figs. 110–114). Transmission electron microscopy (TEM) revealed the formation of spherical assemblies, with a diameter around 200 nm. The size was also confirmed by scanning electron microscopy (SEMs) experiments (Supplementary Fig. 117). The morphology was maintained upon illumination at 650 nm to convert **c-2** to **o-2**. The addition of aniline into **c-2** triggered the decomposition of amphiphiles as a result of C-S bond breaking, and the formation of large aggregates was observed. When the exchange was suppressed

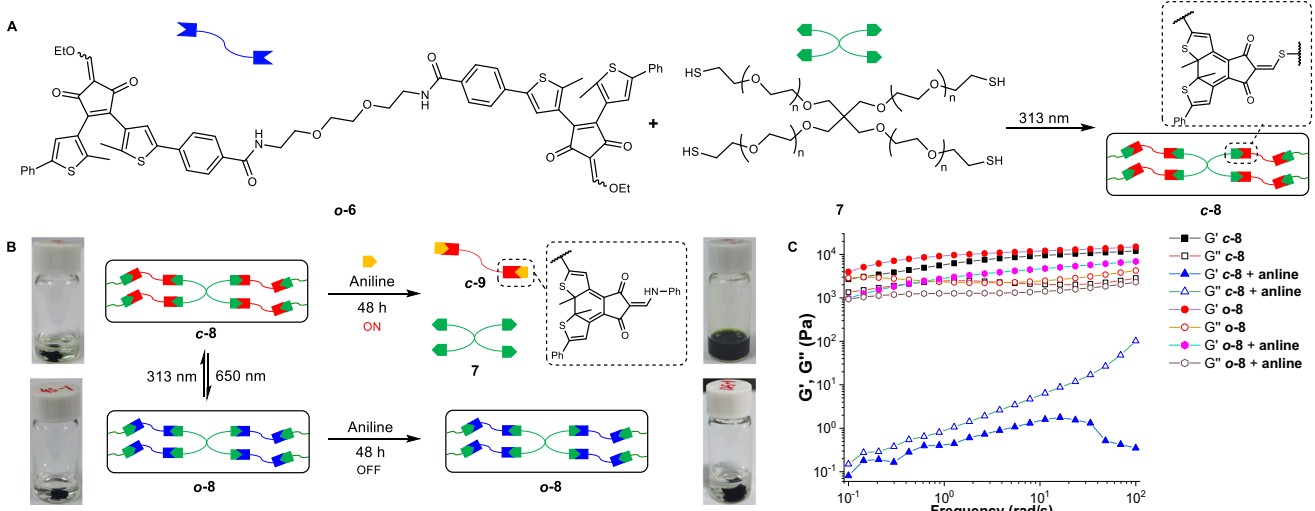

**Fig. 6 | Light-mediated creation/degradation of covalent polymers. A** Light-induced construction of polymer *o-8*/*c-8* from *o-6*/*c-6* and **7**, followed by the degradation **B** with aniline in acetonitrile. Frequency sweep curves of *o-8*/*c-8* upon treatment with aniline in oscillatory rheology analysis (1% strain) are shown (**C**).

with *o-2* the disassociation was accordingly turned off (Supplementary Figs. 115 and 116).

Finally, the regulation of polymers was explored by taking advantage of photoswitchable DCC. The creation of degradable, recyclable polymers is attracting significant attention toward the goal of improved sustainability[67–69]. A bifunctional Michael acceptor (**6**, Fig. 6A and Supplementary Figs. 4–7) was synthesized and characterized. Upon irradiation of *o-6* and tetra-functional thiol crosslinker (**7**) at 313 nm for 2 h, polymerization occurred, creating *c-8* (Fig. 6A and Supplementary Fig. 118). However, no apparent reaction between *o-6* and **7** was found after 7 days (Supplementary Fig. 119). Photocycloreversion of *c-8* at 650 nm afforded *o-8*, and bidirectional switching between *o-8* and *c-8* was feasible (Supplementary Fig. 120). Amine/thiol exchange with aniline enabled the destruction of polymer network (*c-8*) and dissolution after 48 h, confirmed by the observation of aniline derived *c-9* (Fig. 6B). In contrast, the solid-like material of locked *o-8* was maintained and became swollen after two days. Oscillatory rheology (Fig. 6C), NMR analysis (Supplementary Figs. 124 and 125), GPC results (Supplementary Fig. 126), and mass spectra (Supplementary Fig. 127) validated controlled degradation of polymers. For example, the elastic solid feature was evident for *o-8* with or without aniline, possessing a larger storage modulus (G′) than the loss modulus (G″). Differently, upon treatment of *c-8* with aniline G′ was overtaken by G″, echoing the disassembly of polymers (Fig. 6C). The degradation of polymers was further attained with piperidine and diisopropylamine (Supplementary Figs. 128 and 129). As a result, through connecting/disconnecting photoswitchable dynamic covalent linkages light-mediated on-demand construction/degradation of covalent polymers was realized.

In this work, we have developed a versatile platform for light-controlled dynamic covalent conjugate addition-elimination reactions. Photoswitching between open-ring and closed-ring forms of dithienylethene enables the regulation of the reactivity of coupled Michael acceptors, toggling off and on the dynamic exchange of a broad range of thiol and amine nucleophiles for covalent connection and disconnection. Mechanistic studies revealed that the variation of antiaromaticity in the transition states and enol intermediates of conjugate addition-elimination plays a critical role in photoactivated dynamic covalent reactions. To show the potential the utility in different operating environments was demonstrated with photoaddressed modification of surfaces, regulation of amphiphilic assemblies, and creation/degradation of covalent polymers. Due to the mild reaction conditions and readily available reagents the current platform makes a valuable addition to light-mediated click and clip chemistry. The toolbox for spatiotemporal manipulation of dynamic covalent bonds should provide opportunities for many aspects, such as molecular assemblies, targeted delivery, intelligent materials, and sustainable chemistry.

## Methods

### General

¹H NMR and ¹³C NMR spectra were recorded on a 400 MHz Bruker Biospin avance III spectrometer. Deuterated reagents for characterization and in situ reactions were purchased from Sigma-Aldrich Chemical Co. and Cambridge Isotope Laboratories, Inc. (purity ≥99.9%). All other reagents were obtained from commercial sources and were used without further purification, unless indicated otherwise. The chemical shifts (δ) for ¹H NMR spectra, given in ppm, are referenced to the residual proton signal of the deuterated solvent. Mass spectra were recorded on a Bruker IMPACT-II or Thermo-Scientific LCQ Fleet spectrometer. Crystallographic data was collected on a Mercury single crystal diffractometer at room temperature. The structures were solved with direct methods by using OlexSys or SHELXS-97 and refined with the full-matrix least-squares technique based on F2 by using the OlexSys or SHELXL-97. The UV-Vis spectra were recorded on a Perkin-Elmer Lambda 365 spectrometer. TEM images were obtained on a Tecnai F20 Field Emission transmission electron microscope. SEM images were obtained on a SU-8010 Field Emission scanning electron microscope. The UV and visible light irradiation experiments were carried out on a CEL − HXF 300 xenon lamp with bandpass filters at 313 ± 10 nm (2.6 W) and 650 ± 10 nm (16.6 W), respectively.

### Dynamic covalent reactions

Dynamic covalent reactions (DCRs) were performed in situ in CD₃CN at room temperature without isolation and purification. To a stirred solution of Michael acceptor (∼5 mM, 1.0 equiv.) in CD₃CN (0.5 mL), was added one mononucleophile (RSH (3.0 equiv.), RNH₂ (1.5 equiv.), or R₁R₂NH (1.5 equiv.)). The reaction was tracked and characterized by ¹H NMR and ESI mass spectral analysis. See specific conditions in figure captions of the main text or supplementary information if necessary.

## Surface experiments

Glass slides were cleaned for 12 h (each side) in a Piranha solution (15 mL 30 wt% $H_2O_2$ and 35 mL conc. sulfuric acid). The glass slides were rinsed with water and then sonicated in deionized water for 2 min (repeated 2 times). Glass slides were then placed in a vacuum oven for 12 h at 120 °C to fully dry. Dry glass slides were submerged in 10 v/v% solution of 3-(mercaptopropyl)trimethoxysilane (MTS) in dry toluene (~20 h). The slides were then rinsed with toluene and sonicated in toluene for 2 min (repeated 2 times). The slides were placed in a 120 °C vacuum oven for 1 h to anneal. The slides were then rinsed with toluene and deionized water to remove any additional MTS. Contact angle of ultrapure water: 68° ± 1.

To examine surface functionalization by using the reaction of **1** with attached thiol, static contact angle analysis was performed following the procedure. MTS -functionalized slides were subjected to six conditions: a) submerge the slide in acetonitrile as a control for 12 h; b) submerge the slide in a solution of *o*-**1** in acetonitrile (20 mM) for 12 h; c) submerge the slide in a solution of *c*-**1** in acetonitrile (20 mM) for 12 h; d) reaction-modified slide in the step c was placed under 650 nm light for 2 h; e) reaction-modified slide in the step c was submerged in a solution of aniline in acetonitrile (100 mM for 30 min; f) reaction-modified slide in the step d was submerged in a solution of aniline in acetonitrile (100 mM) for 30 min. Each slide was washed with acetonitrile and allowed to dry before analysis. Based on the contact angle analysis, the contact angle of the thiol-modified glass slide did not change after steps a and b, while the contact angle of the glass slide increased after steps c and d. After step e the thiol surface was restored. On the contrary, the surface contact angle remained unchanged after step f. Therefore, light-controlled modification of the solid surface was achieved.

## Light-controlled amphiphilic assemblies

The hydrophilic chain 2,5,8,11-tetraoxatridecane-13-thiol (PEGSH, **5**) was prepared according to the literature[70]. The creation of the amphiphiles was carried out according to the following general procedure. To a solution of *o*-**1** (1.24 mg, 0.0025 mmol) or *c*-**1** (1.24 mg, 0.0025 mmol) in $CD_3CN$ (0.5 mL) was added PEGSH (3 equiv.). The reaction was run for 12 h and monitored by $^1H$ NMR. Compound *c*-**1** completely converted to *c*-**2** (PEGSH) at the end of the reaction, while *o*-**1** did not. Distilled water (5 mL) was slowly dropped into a $CD_3CN$ solution of *c*-**2** (PEGSH) in a glass vial followed by sonication. The mixture was diluted 20 times with a solution of distilled water and MeCN (9:1) and used for SEM and TEM imaging. The solution of *o*-**2** (PEGSH) was created by irradiation of *c*-**2** (PEGSH) at 650 nm for 2 h. The assemblies of *o*-**2** (PEGSH) and other control samples were prepared according to general procedure.

## Light-controlled regulation of polymers

Polymer **8** was prepared and characterized in detail. *o*-**6** (6 mg) and 4-arm poly(ethylene glycol) thiol (≥ 95.0%, $M_n$ ca. 5000 Da) (37.5 mg) were dissolved in $CHCl_3$:$CH_3CN$ (1:5, 0.5 mL) and then illuminated under UV light (313 nm, 2 h) to afford polymer *c*-**8** ($M_n = 107364$, $M_w = 208,222$, PDI = 1.94). *c*-**8** was switched to the insoluble polymer *o*-**8** after illumination under visible light (650 nm, 5 h). The solution was transferred to a mold with a diameter of 20 mm and a thickness of 2 mm and dried for 4 h, and the residue was then swollen in water to remove the residual reagent and dried until a constant weight for storage.

Fourier transform infrared (FTIR) spectra were recorded at room temperature on a Bruker VERTEX70 spectrometer equipped with an attenuated total reflectance (ATR) accessory for direct measurement on starting materials and polymers. Sampling was carried out over 4000–400 $cm^{-1}$. Gel permeation chromatography (GPC) was run on a Waters 1515 or 1525 high performance liquid chromatography instrument equipped with a 2414 refractive index detector and a set of Styragel mixed-C columns. The oven temperature was set at 40 °C. DMF was used as a mobile phase, and the flow rate was 1.0 mL/min. Thermogravimetric analysis (TGA) was obtained on a Netzsch TG 209 *F1 S4Libra*® at a heating rate of 10 °C/min under nitrogen atmosphere. Differential scanning calorimetry (DSC) analyses were performed on a Netzsch DSC 214 Polyma under nitrogen atmosphere at a heating rate of 5 °C/min with an empty aluminum pan as a reference. Tensile tests were conducted on a Shimadzu AG-X Plus 100KN instrument. For tensile tests samples with dimensions of 20 × 3 × 2 mm were used. The rheological properties of *c*-**8** and *o*-**8** were evaluated by an Anton paar rheometer (MCR302). Briefly, the storage modulus (G′) and loss modulus (G″) were measured with the angular frequency from 0.1 to 100 rad/s. All measurements were conducted under constant humidity (60%) and temperature (25 °C).

## Computations

Density functional theory (DFT) calculations were performed by using Gaussian 09 packages. The method and basis set of M06-2X-D3/def2-SVPP with an ultrafine integration grid were employed for the optimization and frequency analysis. The polarizable continuum model (PCM) was included for acetonitrile. By frequency analysis the number of imaginary frequencies for minima and transition state is 0 and 1, respectively. Single point energy calculations were further performed on the pre-optimized geometries at M06-2X-D3/def2-TZVPP level. The nucleus-independent chemical shift (NICS) calculation was conducted at M06-2X/def2-TZVPP level.

## Data availability

The data supporting the findings of this study are provided in the Supplementary Information. The X-ray crystallographic coordinates for structures reported in this study have been deposited at The Cambridge Crystallographic Data Centre (CCDC) under deposition number 2179459 for *o*-**3**. These data can be obtained free of charge from The Cambridge Crystallographic Data Centre at www.ccdc.cam. ac.uk/data_request/cif. Molecular coordinates of calculated structures and crystal structure of *o*-**3** are provided as files Supplementary Data 1 and Supplementary Data 2, respectively. All other data are available from the authors on request.

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

## Acknowledgements

We thank the National Natural Science Foundation of China (92156010, L.Y.; 22071247, L.Y.; 22101283, H.Z.; 22101284, H.Y.), the Strategic Priority Research Program (XDB20000000, L.Y.), the Key Research Program of Frontier Sciences (QYZDB-SSW-SLH030, L.Y.) of the Chinese Academy of Sciences, Natural Science Foundation of Fujian Province (2020J06035, L.Y.), and Fujian Science & Technology Innovation Laboratory for Optoelectronic Information of China (2021ZR112, L.Y.) for funding.

## Author contributions

L.Y. conceived of the idea and directed the research. H.L. designed the protocols, performed the experiments, and analyzed the data. H.Y. conducted density functional theory calculations. M.Z. and Z.L. participated in organic synthesis. H.Z. contributed to polymer characterization. L.Y. wrote the manuscript. All authors discussed the results and revised the manuscript.

## Competing interests

The authors declare the following competing interests: A patent application (China Patent Application 2022109484438) was filed by L.Y., H.L., H.Y., M.Z., and H.Z. of Fujian Institute of the Research on the Structure of Matter on photoswitchable compounds and associated reactivity described in this work. L.Y., H.L., H.Y., M.Z., and H.Z. declare no other competing interests. Z.L. declares no competing interest.
