## [Peer Review File · Nature Communications]

REVIEWER COMMENTS

Reviewer #1 (Remarks to the Author):

This work reports a reversible Michael addition process that is controlled by a photo-driven pericyclic reaction. Upon photo irradiation, the dithienylethene-based electrophile can switch between open vs. closed form, which leads to the antiaromatic vs. nonaromatic intermediate, respectively, thus changing its reactivity dramatically. The bond exchange occurs much faster in the closed form. The reaction mechanism was studied using the computation calculation. The potential applications of such reversible reaction towards surface modification, self-assembly, and polymer degradation were also discussed. This is an interesting work, but the authors need to address the following questions before it is subjected to further consideration:

1. One major concern is about the small molecule model reactions. In Fig 3, the authors gave four examples to prove that the open-form pathway is much slower than the closed-form pathway. Similarly, Table 1 gave the scope of the reaction substrates. However, the substrate scope seems limited. Specifically,

- Can alkoxy group be substituted by alcohols? Can -SR be substituted by thiols?
- Similarly, can -SR be substituted by -NR₁R₂, or -OR? Can -NR₁R₂ be substituted by -SR or -OR?
- Why did the authors use 3 equivalents of thiol, but only 1.5 equivalents of amines?

2. The authors used the DFT calculations to support the notion that the open-form path way is slower because more activation energy is required. It is highly recommended that the activation energy be measured experimentally, which would be more convincing.

Reviewer #2 (Remarks to the Author):

The manuscript by You and coworkers describes a new photoswitchable covalent conjugate addition-elimination reactions and its use in modification of surface wettability, regulation of amphiphilic assemblies and degradation of covalent polymers. The strategy that the reactivity of Michael reactions was tuned through closed/open-ring forms of dithienylethene is a routine step forward based on their published research work (J. Am. Chem. Soc. 2021, 143, 20368–20376, which

is more elaborated therein) and would potentially be of interest to readers studying on dynamic covalent chemistry (DCC) for fabrication on surface or polymeric solidification. However, the novelty of this research work is still insufficiently to show the advantages on high spatiotemporal resolution of the phototriggered click and clip reaction. The applications of this photo-tuned Michael addition/cleavage reaction were quite rough, which is of no valuable advance based on their published work. Therefore, I do not recommend publication of this work unless significant progresses can be accomplished.

In addition, I have the following suggestions:

1. Please double-check for the correct references, such as citation entries: 12, 13, 46 and 64.

2. The use of description "on/off" in Figure 1b could be easily mistaken as the on/off state of the photoswitch moiety rather than whether the reaction can be proceeded. Moreover, this photo-control of the Micheal replacing reaction is paradoxically written as "fast/slow" in Figure 2. The word "reversible" appears several times in the manuscript, and I believe that the reversibility is referring to the reversible substitution from "YR2" to "XR1" on the dithienylethene framework as shown in Figure 1b, but unfortunately this concept is completely not realized in the manuscript.

3. What is the luminous power at 313 nm and 650 nm output from the CEL-HXF 300 Xenon lamp used by the authors? I suppose the photo-switching of DTE should be very fast. In fact, it took only 15 s to reach the PSS at 313 nm irradiation shown in Figure S23 in the SI. Why did it take 1.5 hours to achieve the PSS described in the manuscript? Similarly, the PSS could be achieved via irradiation at 650 nm within 12 minutes in the SI, so why did it take 2 hours to complete described in the manuscript? By the way, the absorbance should be a unitless quantity in Figure S23.

4. Theoretically, c-1 should be completely converted after the 650 nm irradiation, but the authors described the presence of 2% o-1 in Figure S22Ac. but I can hardly see the proton signal of c-1 in Figure 2Bc. Accordingly, are the ^1H NMR in Figure 2Bc and the ^1H NMR in Figure S22Ac the same spectrum? Do the authors have the ^1H NMR spectra with the integration of each proton peak? In Figure 2B, I don't believe the authors were able to obtain a ^1H NMR spectrum of a pure solution of c-1. Therefore, does the red spectral line for compound c-1 in the UV-Vis spectrum represent absorbance at the PSS or the extrapolated spectrum for c-1?

5. Could the second-order reaction rate be measured as a specified value rather than estimated to be enhanced by a factor of 104.

6. This manuscript does not contain any quotation for the S63-S67 in the SI.
7. Please check the structure of c-TS2 and o-TS2 in Figure 4.
8. There should be at least three or more repetitions of experiment for providing a reliable supporting on the contact angle results in Figure 5.
9. I'm not quite sure why there is no color change during the irradiation for the polymeric cross-linking since the photo-switching of dithienylethene is photochromic. In Figure 6B, we can see the solid-like material of o-8 became swollen in the aniline solution accompanied by partial dissolution of compound o-6 in the solvent (solution coloration). Is this coloration caused by incomplete photo-conversion of c-8 to o-8? If so, will the coloration still be observed when the solution is discarded, the solid is then washed with CH₃CN, and a new batch of aniline solution is added?
10. Why were aniline chosen as the nucleophilic reagent in the recyclable polymeric cross-linking experiment? What about the reaction rate when replacing with c-2(1-propanethiol) or diisopropylamine or piperidine?
11. Did all the nucleophilic reactions take place under dark environment? Will the Micheal replacing reaction be accelerated if the nucleophilic reagent is added followed by a continuous 313 nm illumination? Similarly, can polymer c-8 be degraded more rapidly in aniline solutions, when promoted by exposure to light, e.g., sunshine?
12. I strongly endorse that phototriggered click and clip reactions can endow chemical processes with a high spatiotemporal resolution and sustainability. However, the high spatial resolution was not demonstrated at all throughout this work. Is it possible to achieve patterned surface modification on glass at the micron level as in citations 65 and 66, or achieve controlled degradation of polymers for 3D printing of microstructures?

Reviewer #3 (Remarks to the Author):

The objective of this work was to present a photo-switchable reaction that could be added to the click/clip reaction toolbox. This versatile technique utilizes photo-controlled Michael-type reactions to create a stronger electrophile after closure of a diarylethene photoswitch, in the open form it is less electrophilic. Therefore, this photo-controlled reaction acts as an “on” and “off” switch to control reactivity. This process is driven by switching between 4 and 14 pi electrons. The mechanism was probed via DFT calculations. Potential applications for this finding were explored. Regulation of molecular assemblies through photo switchable surface modification was achieved in microfluidic chips. Amphiphilic assemblies were tested for versatility. Through connecting and disconnecting covalent linkages, light mediated production of degradable and recyclable polymers was achieved.

I thoroughly enjoyed reading this paper, the photo switchable reaction was intriguing and clearly explained within the paper and punctuated by the figures included. The information was presented in a clear and logical flow. Throughout the paper, findings were well supported. Each claim made was followed up with additional studies and convincing data.

I recommend that the paper should be accepted. The work done was well supported and thoroughly backed up. I have no revisions to suggest.

Questions:

258 in regulation of molecular assemblies – why feasible, what does that really mean were they or were they not able to observe bidirectional switching?

April 29, 2023

Dear Reviewers,

Thank you for reviewing manuscript NCOMMS-22-42849. We have uploaded our revised manuscript entitled “**Photoswitchable Dynamic Conjugate Addition-Elimination Reactions: A New Tool for Light-Mediated Click and Clip Chemistry**” that we wish to have considered for publication as an article in *Nature Communications*.

The reviewers provided very helpful comments, and we have made every effort to address the comments and improve this manuscript (the changes/additions are highlighted in red in a copy of the manuscript as Supporting Information). I have so indicated in my point-by-point response and revision summary, which follows.

Responses to comments by Reviewer 1:

This work reports a reversible Michael addition process that is controlled by a photo-driven pericyclic reaction. Upon photo irradiation, the dithienylethene-based electrophile can switch between open vs. closed form, which leads to the antiaromatic vs. nonaromatic intermediate, respectively, thus changing its reactivity dramatically. The bond exchange occurs much faster in the closed form. The reaction mechanism was studied using the computation calculation. The potential applications of such reversible reaction towards surface modification, self-assembly, and polymer degradation were also discussed. This is an interesting work, but the authors need to address the following questions before it is subjected to further consideration.

Response: Thanks a lot for the reviewer’s comments and suggestions on our manuscript.

Comment: One major concern is about the small molecule model reactions. In Fig 3, the authors gave four examples to prove that the open-form pathway is much slower than the closed-form pathway. Similarly, Table 1 gave the scope of the reaction substrates. However, the substrate scope seems limited. Specifically,

- Can alkoxy group be substituted by alcohols? Can -SR be substituted by thiols?
- Similarly, can -SR be substituted by -NR₁R₂, or -OR? Can -NR₁R₂ be substituted by -SR or -OR?
- Why did the authors use 3 equivalents of thiol, but only 1.5 equivalents of amines?

Response: Thanks a lot for the reviewer’s comments. To expand the scope of the substrates, additional exchange experiments were performed:

- The reaction of **c-1** with 1-propanol (3 eq) gave desired **c-1**(1-propanol) through alcohol exchange, while **o-1** remained virtually intact after three weeks (Table 1 entry 1 and Figures S25-S27). The exchange reactions between thiols have been showed with 1-propanethiol and benzythiol (Table 1 entry 7 and Figures S47-S49).
- The thiol incorporated into **c-2**(1-propanethiol) can be substituted by primary (Table 1 entries 14-15 and Figures S71-S75) or secondary amines, such as diisopropylamine and piperidine with high efficiency (Table 1 entries 19-20 and Figures S86-S89). Owing

to the order of nucleophilicity (amines > thiols > alcohols), it is generally infeasible to exchange amines with alcohols/thiols and to exchange thiols with alcohols.

- Due to the higher nucleophilicity of amines, 1.5 equivalents of amines were used in addition-elimination reactions, as compared to 3 equivalents of alcohols/thiols.

Comment: The authors used the DFT calculations to support the notion that the open-form path way is slower because more activation energy is required. It is highly recommended that the activation energy be measured experimentally, which would be more convincing.

Response: We appreciate it very much for the suggestion. The activation energies of reactions **c-1/o-1** with thiol were measured experimentally via pseudo-first order kinetics, and the related data have been added in the revised manuscript and supporting information (Figures 4B and S98-S99). The difference in measured reaction barriers of ring-closed and ring-open forms was found to be 5.31 kcal•mol⁻¹, in consistent with calculated value (5.70 kcal•mol⁻¹). In addition, the rate constants of reactions of **c-1/o-1** with aniline at room temperature were measured, modulating the rate over 4 orders of magnitude (Figure S100).

Responses to comments by Reviewer 2:

The manuscript by You and coworkers describes a new photoswitchable covalent conjugate addition-elimination reactions and its use in modification of surface wettability, regulation of amphiphilic assemblies and degradation of covalent polymers. The strategy that the reactivity of Michael reactions was tuned through closed/opening forms of dithienylethene is a routine step forward based on their published research work (*J. Am. Chem. Soc.* **2021**, *143*, 20368-20376, which is more elaborated therein) and would potentially be of interest to readers studying on dynamic covalent chemistry (DCC) for fabrication on surface or polymeric solidification. However, the novelty of this research work is still insufficiently to show the advantages on high spatiotemporal resolution of the phototriggered click and clip reaction. The applications of this photo-tuned Michael addition/cleavage reaction were quite rough, which is of no valuable advance based on their published work. Therefore, I do not recommend publication of this work unless significant progresses can be accomplished.

Response: Thanks a lot for the reviewer's comments and suggestions on our manuscript.

The development of light-induced dynamic chemistry and materials is of both scientific importance and practical interest, as light represents one green and non-invasive energy source. To enrich the toolbox of photoswitchable dynamic covalent reactions, we have presented the modulation of conjugate addition-elimination reactions, one important class of click/clip reactions, by light in this work. The current research results are novel as compared to our previously published work (*J. Am. Chem. Soc.* **2021**, *143*, 20368-20376, ref. 30) for three points: 1. The reaction type and mechanism are different: the current work focuses on Michael-type reactions with associative exchange mechanism, while the previous work describes neighboring carboxyl assisted

acetal/thioacetal/aminal chemistry and the exchange can proceed via dissociative mechanism; 2. The reaction conditions are different: the current work runs exchange reactions of alcohols/thiols in neutral conditions, while acid is required to facilitate reactions of alcohols/thiols in the previous work; 3. The reaction scopes are different: the exchange reactions of Michael acceptors with different amino acids can be performed in aqueous solution (Table 1 entries 6, 13, 16 and Figures S44-S46, S68-S70, and S76-S78), showing the potential in physiological environment. As a result, considering the broad scope and mild condition the current work makes a valuable addition into the toolbox of light-mediated dynamic click/clip chemistry.

Moreover, to demonstrate the impact we described versatile utility of photoinduced regulation of surface wettability, amphiphilic assemblies, and degradable covalent polymers, and we further performed light-controlled patterning to show the spatiotemporal resolution, as detailed in the response later. These results would enable future applications of photoswitchable reversible covalent linkages in molecular assemblies, surface engineering, smart materials, biomedicine, and improved sustainability. Therefore, we feel that the current work has the novelty and significance required for publication in *Nature Communications*, as other two reviewers agreed.

Comment: Please double-check for the correct references, such as citation entries: 12, 13, 46 and 64.

Response: Thanks a lot for careful checking. We have scrutinized the reference list and corrected these errors.

Comment: The use of description “on/off” in Figure 1b could be easily mistaken as the on/off state of the photoswitch moiety rather than whether the reaction can be proceeded. Moreover, this photo-control of the Michael replacing reaction is paradoxically written as “fast/slow” in Figure 2. The word “reversible” appears several times in the manuscript, and I believe that the reversibility is referring to the reversible substitution from “YR₂” to “XR₁” on the dithienylethene framework as shown in Figure 1b, but unfortunately this concept is completely not realized in the manuscript.

Response: We fully agree with the reviewer and made the change in Figure 2 and the main text to use “fast/slow” to reflect rate acceleration in the photochemical ring-closed form. Our use of the word “reversible” was in reference to dynamic covalent reactions, which are reversible in nature. The outcomes of conjugate addition-elimination reactions must follow thermodynamics and hence the order of nucleophilicity of exchange partners (HYR₂ and HXR₁), as in the response to Reviewer 1. To avoid misunderstandings we have rephrased the relevant sentences in the main text.

Comment: What is the luminous power at 313 nm and 650 nm output from the CEL-HXF 300 Xenon lamp used by the authors? I suppose the photo-switching of DTE should be very fast. In fact, it took only 15 s to reach the PSS at 313 nm irradiation shown in Figure S23 in the SI. Why did it take 1.5 hours to achieve the PSS described in the manuscript? Similarly, the PSS could be achieved via irradiation at 650 nm within

12 minutes in the SI, so why did it take 2 hours to complete described in the manuscript? By the way, the absorbance should be a unitless quantity in Figure S23.

Response: Thanks for the reviewer's comments. The luminous power is 2.6 W at 313 nm and 16.6 W at 650 nm. The longer irradiation time in Figure 2 is due to significantly higher concentration for samples in NMR analysis than UV-vis measurements (Figure S23). We have made the change in Figure S23 to ensure the absorbance a unitless quantity.

Comment: Theoretically, **c-1** should be completely converted after the 650 nm irradiation, but the authors described the presence of 2% **o-1** in Figure S22Ac. but I can hardly see the proton signal of **c-1** in Figure 2Bc. Accordingly, are the ¹H NMR in Figure 2Bc and the ¹H NMR in Figure S22Ac the same spectrum? Do the authors have the ¹H NMR spectra with the integration of each proton peak? In Figure 2B, I don't believe the authors were able to obtain a ¹H NMR spectrum of a pure solution of **c-1**. Therefore, does the red spectral line for compound **c-1** in the UV-Vis spectrum represent absorbance at the PSS or the extrapolated spectrum for **c-1**?

Response: Thanks a lot for pointing out this issue. We rephrased the description of the irradiation experiment to show quantitative conversion for photocycloreversion at 650 nm and added the integration of associated proton peaks of the ¹H NMR spectra in the revised supporting information (Figure S22). We have corrected the description in the caption of Figure S22. The red spectral line for compound **c-1** in the UV-Vis spectrum represents the absorbance at the PSS.

Comment: Could the second-order reaction rate be measured as a specified value rather than estimated to be enhanced by a factor of 10⁴.

Response: We thank the reviewer for this suggestion. We have measured rate constants and reaction barriers, as also suggested by Reviewer 1.

Comment: This manuscript does not contain any quotation for the S63-S67 in the SI

Response: Thank the reviewer for careful checking. We have added the missing quotation in the revised manuscript.

Comment: Please check the structure of **c-TS2** and **o-TS2** in Figure 4.

Response: Thanks for the correction. We have checked the structure of **c-TS2** and **o-TS2** in Figure 4 and made the change.

Comment: There should be at least three or more repetitions of experiment for providing a reliable supporting on the contact angle results in Figure 5.

Response: Thanks for the kind suggestion. We have repeated the contact angle measurement for three times, and the results were repeatable.

Comment: I'm not quite sure why there is no color change during the irradiation for the polymeric cross-linking since the photo-switching of dithienylethene is photochromic. In Figure 6B, we can see the solid-like material of **o-8** became swollen in the aniline

solution accompanied by partial dissolution of compound **o-6** in the solvent (solution coloration). Is this coloration caused by incomplete photo-conversion of **c-8** to **o-8**? If so, will the coloration still be observed when the solution is discarded, the solid is then washed with CH₃CN, and a new batch of aniline solution is added?

Response: We appreciate it very much for the suggestion. The dark green color of polymer **o-8** after concentration makes it difficult to visually distinguish from **c-8**. We agree that the coloration of the solution was caused by the incomplete photo-conversion of **c-8** to **o-8**. After the solid was washed with CH₃CN followed by adding a new batch of aniline solution, the solution remained colorless after another 48 h (Figure S119).

Comment: Why were aniline chosen as the nucleophilic reagent in the recyclable polymeric cross-linking experiment? What about the reaction rate when replacing with **c-2**(1-propanethiol) or diisopropylamine or piperidine?

Response: Thanks the reviewer for suggestions. From the model reaction of small molecules (Table 1) it can be concluded that **c-2** is much more reactive with aniline than **o-2**, and hence, aniline was employed for the degradation of polymers. We have also conducted the reactions of **c-8/o-8** with diisopropylamine and piperidine, and the results have been added in the revised supporting information (Figures S121-122).

Comment: Did all the nucleophilic reactions take place under dark environment? Will the Micheal replacing reaction be accelerated if the nucleophilic reagent is added followed by a continuous 313 nm illumination? Similarly, can polymer **c-8** be degraded more rapidly in aniline solutions, when promoted by exposure to light, e.g., sunshine?

Response: Thanks for reviewer's comments. We performed the dynamic reactions in this work in a dark environment to avoid potential side effect of UV light. As suggested by the reviewer, the reaction of **o-1** with 1-propanethiol was run under 313 nm illumination. After PSS was reached, the reaction proceeded faster than **c-1** with 1-propanethiol in the dark (Figure S31-S32). In addition, the less degradation time of polymer **c-8** was found under UV light (Figure S119).

Comment: I strongly endorse that phototriggered click and clip reactions can endow chemical processes with a high spatiotemporal resolution and sustainability. However, the high spatial resolution was not demonstrated at all throughout this work. Is it possible to achieve patterned surface modification on glass at the micron level as in citations 65 and 66, or achieve controlled degradation of polymers for 3D printing of microstructures?

Response: We appreciate it very much for the suggestion. To demonstrate the spatial resolution of light, the modification of patterning was shown with color changes of **1** loaded filter papers in response to thiol vapor (Figure 5B).

Responses to comments by Reviewer 3:

The objective of this work was to present a photo-switchable reaction that could be added to the click/clip reaction toolbox. This versatile technique utilizes photo-

controlled Michael-type reactions to create a stronger electrophile after closure of a diarylethene photoswitch, in the open form it is less electrophilic. Therefore, this photo-controlled reaction acts as an “on” and “off” switch to control reactivity. This process is driven by switching between 4 and 14 pi electrons. The mechanism was probed via DFT calculations. Potential applications for this finding were explored. Regulation of molecular assemblies through photo switchable surface modification was achieved in microfluidic chips. Amphiphilic assemblies were tested for versatility. Through connecting and disconnecting covalent linkages, light mediated production of degradable and recyclable polymers was achieved.

I thoroughly enjoyed reading this paper, the photo switchable reaction was intriguing and clearly explained within the paper and punctuated by the figures included. The information was presented in a clear and logical flow. Throughout the paper, findings were well supported. Each claim made was followed up with additional studies and convincing data.

I recommend that the paper should be accepted. The work done was well supported and thoroughly backed up. I have no revisions to suggest.

Response: Thanks a lot for the reviewer’s comments and suggestions on our manuscript.

Comment: Line 258 in regulation of molecular assemblies - why feasible, what does that really mean were they or were they not able to observe bidirectional switching?

Response: Thanks for reviewer’s comments. Since the diarylethene photoswitch is capable of bidirectional control under two wavelengths of light, light-induced interconversion between polymers **c-8** and **o-8** in the solution of CH₃CN was realized (Figure S113). By using this strategy light-mediated construction/degradation of polymers was further demonstrated.

We believe that the revised manuscript is improved, and we hope that the manuscript is now acceptable for publication.

Thank you for your acceptance of our work and best personal regards.

Sincerely,

Lei You
Professor
Fujian Institute of Research on the Structure of Matter
Chinese Academy of Sciences
Fuzhou China, 350002

REVIEWERS' COMMENTS

Reviewer #1 (Remarks to the Author):

< In private comments to the Editorial office, the reviewer expressed that the manuscript is ready for publication as-is. >

Reviewer #2 (Remarks to the Author):

The manuscript by You and coworkers describes a new photoswitchable covalent conjugate addition-elimination reactions and its use in modification of surface wettability, regulation of amphiphilic assemblies and degradation of covalent polymers. The strategy that the reactivity of Michael reactions was tuned through closed/open-ring forms of dithienylethene is a big step forward and would be of interest to the readers of Natural Communication. After revision, the authors responded to my concerns and suggestion pretty well.

Since the authors were able to address the spatially controlled application of the phototriggered click and clip reaction, I am supportive to the publication of the manuscript in current version.